



# AsiaRiceMap10m: high-resolution annual paddy rice maps for Southeast and Northeast Asia from 2017 to 2019

Jichong Han[1], Zhao Zhang[1], Yuchuan Luo[1], Juan Cao[1], Liangliang Zhang[1], Fei Cheng[1], Huimin Zhuang[1], Jing Zhang[1]

[1]State Key Laboratory of Earth Surface Processes and Resource Ecology/ MoE Key Laboratory of Environmental Change and Natural Hazards, Faculty of Geographical Science, Beijing Normal University, Beijing 100875, China

*Correspondence to*: Zhao Zhang (sunny_zhang@bnu.edu.cn)

**Abstract.** An accurate paddy rice map is crucial for ensuring food security, particularly for Southeast and Northeast Asia. MODIS satellite data are useful for mapping paddy rice at continental scales but have a mixed pixel problem caused by the
coarse spatial resolution. To reduce the mixed pixels, we designed a rule-based method for mapping paddy rice by integrating time-series Sentinel-1 and MODIS data. We demonstrated the method by generating annual paddy rice maps for Southeast and Northeast Asia in 2017–2019 (AsiaRiceMap10m). We compared the resultant paddy rice maps with available agricultural statistics at subnational levels and existing rice maps for some countries. The results demonstrated that the linear coefficient of determination ($R^2$) between our paddy rice maps and agricultural statistics ranged from 0.80 to 0.97. The
paddy rice planting areas in 2017 were spatially consistent with the existing maps in Vietnam ($R^2 = 0.93$) and Northeast China ($R^2 = 0.99$). The spatial distribution of the 2017–2019 composite paddy rice map was consistent with that of the rice map from the International Rice Research Institute. The paddy rice planting area may have been underestimated in the region in which the flooding signal was not strong. The dataset is useful for water resource management, rice growth, and yield monitoring. The full product is publicly available at https://doi.org/10.17632/j34b3jsvr9.1 (Han et al., 2021a). Find small
examples here (https://doi.org/10.17632/cnc3tkbwcm.1) (Han et al., 2021b).

## 1 Introduction

Rice is one of the main food sources, accounting for approximately 12% of the global cropland area (Zhang et al., 2018; Singha et al., 2019). Approximately 90% of the world's rice is produced in Asian countries (Chen et al., 2012; Yeom et al., 2021). Rice provides food for over 50% of the world's population (Minasny et al., 2019). The consumption of rice increases
as the world's population increases. Additionally, approximately one-tenth of $CH_4$ emissions in the atmosphere come from methane emissions from rice paddies (Ehhalt et al., 2001; Xin et al., 2017; Zhang et al., 2020). Rice agriculture is significant in food security, water resource security, disease transmission, and environmental sustainability (Clauss et al., 2018b; Li et al., 2020; Park et al., 2018). An accurate planting area and spatial distribution information are the basis for monitoring paddy rice growth and predicting yield. However, few spatial maps of paddy fields at continental scales exist (Li et al., 2020;
Singha et al., 2019). Therefore, it is necessary to produce a paddy rice map dataset with high spatial resolution.



Many methods for mapping rice have been developed based on different remote sensing data, including (1) machine learning classifiers (e.g., random forest and support vector machines), (2) phenology-based classifiers, (3) rule-based algorithms, and (4) the time-series algorithm approach (Dong et al., 2016b; Bazzi et al., 2019; Dong et al., 2016a; Dong and Xiao, 2016; Luo et al., 2020b, a; Nelson et al., 2014; Phung et al., 2020; Minasny et al., 2019; Shew and Ghosh, 2019; Xiao et al., 2006; Zhan et al., 2021). Satellite image sources include MODIS, Landsat, Sentinel, RADARSAT, and PALSAR (Dong and Xiao, 2016; Shao et al., 2001; Singha et al., 2019; Zhou et al., 2016). Many studies have demonstrated that phenology-based classifiers using MODIS data are useful for mapping paddy rice at continental scales (Dong et al., 2016b; Xiao et al., 2006; Zhang et al., 2020). The transplanting period of rice is a distinct characteristic used for distinguishing rice from other crops or land-use types. For example, Xiao et al. (2006) mapped paddy rice at continental scales (South Asia and Southeast Asia (SE Asia)) using the phenological characteristics in the period of flooding/transplanting. Additionally, this method was successfully applied in other large regions (Xin et al., 2020; Zhang et al., 2017, 2020). The International Rice Research Institute (IRRI) extracted the distribution of paddy rice for Asia (Nelson and Gumma, 2015). However, the paddy rice maps generated using MODIS data contain a large number of mixed pixels caused by the coarse spatial resolution (500 m) (Dong et al., 2015, 2016b; Shew and Ghosh, 2019), particularly in hilly areas (Liu et al., 2019). The mixed land cover types within MODIS pixels can affect the accuracy of the rice map (Sun et al., 2009). Fine spatial resolution images, including Landsat TM/ETM +/OLI, HJ, and Sentinel-2 images, are also used for mapping paddy rice. Some previous studies have shown that rice maps generated from Landsat images have relatively high accuracy (Dong et al., 2016a; Torbick et al., 2017). However, they are only suitable for relatively small study areas in which cloud cover is minimal, and not for continental scales (Ramadhani et al., 2020; Torbick et al., 2017). In contrast to optical satellite images, synthetic aperture radar (SAR) data are unaffected by clouds (Park et al., 2018). Moreover, SAR data have the special characteristics of backscatter changes during the growth of paddy rice (Bazzi et al., 2019; Clauss et al., 2018b; Nguyen et al., 2016; Phung et al., 2020; Planque et al., 2021). For example, Singha et al. (2019) mapped the rice map for Bangladesh based on Sentinel-1 data and random forest classifiers with good accuracy. Although paddy rice has been mapped in several studies using SAR data, they are still difficult to use widely over large areas because of the lack of a large number of ground truth samples (Clauss et al., 2018b; Minh et al., 2019; Nguyen et al., 2016; Phung et al., 2020; Minasny et al., 2019; Zhan et al., 2021; Zhang et al., 2018). Because the average area of crop fields in many regions in Asia is less than half a hectare (Maclean et al., 2013), it is critical to generate paddy rice maps with higher spatial resolution at continental scales than past efforts with MODIS.

Optical remote sensing images and SAR data have complementary information (Park et al., 2018; Wang et al., 2015). The combination of optical and SAR images can provide opportunities for mapping paddy rice with a few mixed pixels and a high spatial resolution at continental scales. MODIS data have the advantage of high temporal resolution, which reduces cloud problems and provides valuable spectral information for identifying paddy rice. Sentinel-1 SAR data with a high spatial resolution (10 m) provide backscatter information for different land types. Therefore, the integration of MODIS and




SAR images may solve the mixed pixel issue to a great degree and enable the production of more reliable paddy rice maps
than those based only on MODIS images (Dong and Xiao, 2016; Park et al., 2018; Torbick et al., 2010; Wang et al., 2015).
We take advantage of both MODIS and SAR strengths to map paddy rice fields at a large scale.

Thus, we aim to improve the MODIS-based method for mapping paddy rice fields by integrating Sentinel-1 SAR data to
reduce mixed pixel effects. Then we use the method to generate paddy rice maps in 2017–2019 for SE Asia and Northeast
Asia (NE Asia). The map products will be useful for scientific communities and stakeholders for many purposes.

## 2 Materials

### 2.1 Study area

The study areas were NE and SE Asia. NE Asia is composed of Northeast China (Liaoning, Jilin, and Heilongjiang province),
the Democratic People's Republic of Korea, the Republic of Korea, and Japan (Dong et al., 2016b; Yeom et al., 2021). The
main paddy rice-producing regions in NE Asia are concentrated in the plain in Northeast China, the western plain of the
Korean Peninsula, and the alluvial plains around the Japanese islands. In SE Asia, the countries where rice is planted
intensively include Indonesia, Thailand, Vietnam, Myanmar, the Philippines, Malaysia, and Myanmar. SE Asia cultivates
approximately 30% of the world's rice (Bridhikitti and Overcamp, 2012; Huke and Huke, 1997). The dense planting areas of
rice in SE Asia are located in valleys and deltas, such as the Red River Delta in Northern Vietnam and the Mekong Delta in
Southern Vietnam (Clauss et al., 2018a; Phung et al., 2020). The Mekong Delta produces more than half the rice in Vietnam
(Bouvet and Le Toan, 2011). The main rice cropping system in NE Asia is single rice (Dong et al., 2016b). By contrast, three
rice cropping systems are dominant in SE Asia: single rice, double rice, and triple rice (Laborte et al., 2017). Because
climate and crop calendars vary across SE and NE Asia, the study area was classified into eight refined agroecological zones
based on temperature, seasonal precipitation, and farming practices from previous studies (Oliphant et al., 2019; Suepa et al.,
2016). The zones were further subdivided into 41 regions for classification (Figure 1).

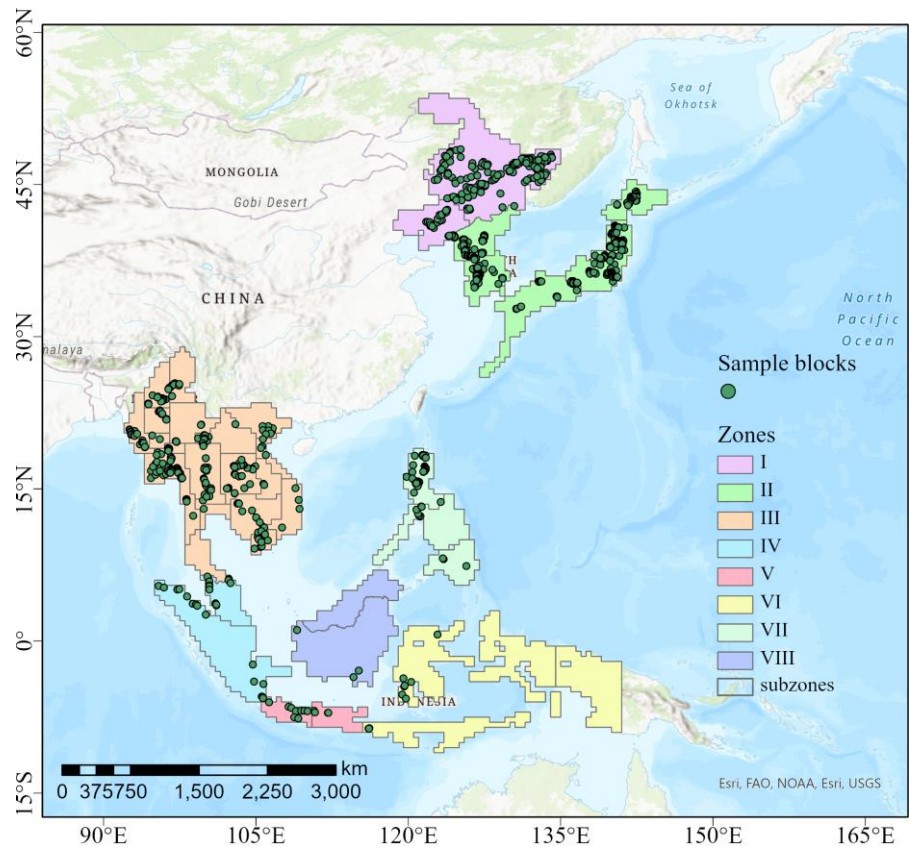

**Figure 1: Agroecological zones and 100 m radius sample blocks in SE and NE Asia.**

## 2.2 Data source

### 2.2.1 Satellite imagery

We acquired the time-series MOD09A1 images from the Google Earth Engine (GEE) data catalog (https://developers.google.com/earth-engine/datasets/, last access: 2021/06/18). The 8-day composite MOD09A1 provides seven surface reflectance bands (red, blue, green, near infrared, and shortwave infrared 1-2) at 500 m spatial resolution. We used the blue band ($\geq 0.2$) to remove cloudy pixels (Chen et al., 2012; Xiao et al., 2006). We projected MODIS data using the WGS1984 coordinate system. Additionally, we collected SAR data with 10 m resolution in interferometric wide swath mode from the Sentinel-1 satellite (Torres et al., 2012). To maximize the frequency of observations, we used the Level 1 ground range detected (GRD) product in ascending and descending orbits. We pre-processed the GRD data with VH and VV polarization that we acquired (e.g., calibration and geocoding) using the Sentinel-1 Toolbox on the GEE (https://developers.google.com/earth-engine/datasets/catalog/COPERNICUS_S1_GRD, last access: 2021/06/18) (Singha et

al., 2019; Liu et al., 2020). Then, we applied a filter with a boxcar kernel (30 × 30) moving to reduce speckle noise (Minasny et al., 2019). However, the rugged terrain and side-looking SAR imaging geometry caused radiometric distortion (Mullissa et al., 2021; Vollrath et al., 2020). To increase the quality of observations, we used the physical reference model (volume) proposed by Vollrath et al. (2020) to make a radiometric slope correction for Sentinel-1 on the GEE. Figure S1 shows an

example of the effect of slope correction in SE Asia using the model. The code can be found at https://github.com/ESA-PhiLab/radiometric-slope-correction (last access: 2021/06/18).

### 2.2.2 Terrain data

We generated digital elevation model (DEM) data from the Shuttle Radar Topography Mission (SRTM) Version 4 (Reuter et al., 2007). The spatial resolution of the DEM was 90 m × 90 m. We acquired the DEM data and calculated the slope map

from the DEM on the GEE platform (Table 1).

### 2.2.3 Forest land

We extracted the forest land mask from the Global PALSAR Forest Map in 2017 (Table 1). The Global PALSAR Forest Map (25-m spatial resolution) was generated by the Japan Aerospace Exploration Agency (JAXA) (Shimada et al., 2014). Pixels with a forest area larger than 0.5 ha and forest covering over 10% of the pixel area were defined as forest pixels

(Shimada et al., 2014).

### 2.2.4 Wetland

We extracted the distribution of wetland from the GlobeLand30 dataset in 2020. GlobeLand30 is available from the National Geomatics Center of China (Table 1). This product at 30-m spatial resolution with high accuracy was generated using Landsat, Chinese HJ-1, and GF-1 satellite images (http://www.globallandcover.com, last access: 2021/06/18) (Chen et al.,

120    2015).

Finally, we resampled all the raster data to 10 m to match the spatial resolution of Sentinel-1.

### 2.2.5 Agricultural statistics

We collected annual rice planting area census data at the subnational level (state, province, city, prefecture, or county) from

the available Statistical Yearbooks of various countries. The agricultural statistics were provided by agricultural statistical offices. The areas in the statistics data were converted into hectares (ha). Detailed information about the collected agricultural statistics in this study is presented in Table 1.





### 2.2.6 Existing rice maps

We collected the existing publicly available rice maps from three sources: (1) the 500 m spatial resolution paddy rice map
with high accuracy in Southern China in 2017 that was generated using the phenology-and pixel-based algorithm from
MODIS data (Xin et al., 2020), (2) the High-Resolution Land Use and Land Cover (HRLULC) map for Vietnam in 2017
(HASHIMOTO et al., 2014) with 10 m spatial resolution generated using multiple remote sensing data, and (3) the 500 m
resolution rice maps of Asia obtained from the IRRI (Nelson and Gumma, 2015), which were mainly derived from MODIS
data. We compared these existing products with our paddy rice maps.



**Table1: Detailed information about the data used in this study.**

| Data type | Data product or country name | Year | Resolution | Descriptions | Data access | Last access (yyyy/mm/dd) |
|---|---|---|---|---|---|---|
| Satellites Imagery | MOD09A1 | 2017-2019 | 500m | Extracting the spectral characteristics and phenology information | https://developers.google.com/earth-engine/datasets/catalog/MODIS_006_MOD09A1 | 2021/06/18 |
| | Sentinel-1 | 2017-2019 | 10m | Extracting the backscatter coefficient characteristics and phenology information | https://developers.google.com/earth-engine/datasets/catalog/COPERNICUS_S1_GRD | 2021/06/18 |
| Terrain data | SRTM V4 | - | 90m | Extracting slope map | https://developers.google.com/earth-engine/datasets/catalog/CGIAR_SRTM90_V4 | 2021/06/18 |
| Existing rice maps | Paddy rice map in Northeast China | 2017 | 500m | Comparing spatial consistency among products | https://figshare.com/s/56f22588ee25330b9d37 | 2021/06/18 |
| | HRLULC map in Vietnam | 2017 | 10m | Comparing of spatial consistency among products | https://www.eorc.jaxa.jp/ALOS/en/lulc/lulc_index.htm | 2021/06/18 |
| | IRRI rice map in Asia | 2000-2012 | 500m | Comparing of spatial consistency among products | http://irri.org/our-work/research/policy-and-markets/mapping | 2021/06/18 |
| Forest land | The Global PALSAR Forest Map | 2017 | 20m | Extracting forest map | https://developers.google.com/earth-engine/datasets/catalog/JAXA_ALOS_PALSAR_YEARLY_FNF | 2021/06/18 |
| Wetlands | Globalland30 | 2020 | 30m | Extracting wetlands map | http://www.globallandcover.com/home_en.html | 2021/06/18 |
| Annual agricultural statistics | Jilin province (China) | 2017-2019 | City scale | Verifying the classification accuracy | http://tjj.jl.gov.cn/tjsj/tjnj/ | 2021/06/18 |
| | Liaoning province (China) | 2017-2019 | City scale | Verifying the classification accuracy | http://tjj.ln.gov.cn/tjsj/sjcx/ndsj/ | 2021/06/18 |
| | Republic of Korea | 2017-2019 | County scale | Verifying the classification accuracy | https://kosis.kr/statisticsList/statisticsListIndex.do?vwcd=MT_ZTITLE&menuId=M_01_01 | 2021/06/18 |
| | Japan | 2017-2019 | Prefecture scale | Verifying the classification accuracy | https://www.stat.go.jp/english/data/nenkan/index.html | 2021/06/18 |
| | Vietnam | 2017-2019 | Province scale | Verifying the classification accuracy | https://www.gso.gov.vn/en/statistical-data/ | 2021/06/18 |
| | Myanmar | 2017-2018 | State scale | Verifying the classification accuracy | https://www.csostat.gov.mm/ | 2021/06/18 |
| | Philippines | 2017-2019 | Province scale | Verifying the classification accuracy. We used the annual area of irrigated rice. | https://openstat.psa.gov.ph/Database/Agriculture-Forestry-Fisheries | 2021/06/18 |



### 2.3 Methodology

#### 2.3.1 Analyzing the characteristics of spectral indices from paddy rice

There are three growing stages for paddy rice: transplanting, growing, and post-harvest periods (Singha et al., 2019).
Flooding signals in the transplanting period are unique characteristics that distinguish paddy rice from other crops (Clauss et al., 2016; Dong et al., 2016b; Sun et al., 2009). The color combination of MODIS images (R/G/B = band7/ band2/ band1) in the transplanting stage of the paddy rice field has a prominent tone (Figure S4). We calculated the Land Surface Water Index (LSWI) and Enhanced Vegetation Index (EVI) for each image:

$$LSWI = \frac{\rho_{NIR} - \rho_{SWIR}}{\rho_{NIR} + \rho_{SWIR}} \tag{1}$$

$$EVI = 2.5 \times \frac{\rho_{NIR} - \rho_{RED}}{\rho_{NIR} + 6 \times \rho_{RED} - 7.5 \times \rho_{BLUE} + 1}, \tag{2}$$

where $\rho_{SWIR}$, $\rho_{NIR}$, $\rho_{RED}$, and $\rho_{BLUE}$ are values of band 6, band 2, band 1, and band 3, respectively. Note that we chose EVI instead of the normalized difference vegetation index (NDVI) for paddy rice identification because NDVI is more sensitive to atmospheric contamination and has a saturation issue (Zhang et al., 2015). Figure 2 shows the standard temporal profile of the EVI and LSWI of different paddy rice planting systems (single, double, and triple) at three typical sites. When LSWI plus 0.05 is larger than EVI, it indicates that the paddy rice is in the transplanting period (the). Both the EVI and LSWI values increase after paddy rice is transplanted. EVI values are higher than LSWI values because the fields are fully covered by the rice canopy. The EVI decreases during the post-harvest period. Both double paddy rice and triple paddy rice have flooding signals. The above phenomena are consistent with previous studies (Dong et al., 2016b; Minh et al., 2019; Shew and Ghosh, 2019; Xiao et al., 2006). Therefore, color gradations and the relationship between EVI and LSWI are useful for extracting phenological information of paddy rice.

#### 2.3.2 Analyzing the characteristics of backscatter coefficients from paddy rice

The backscatter coefficients change as paddy rice grows and develops. Paddy rice fields appear as a black area in the VH image on the transplanting date (Figure S2) because the water (flood) in the transplanting period decreases the VH backscatter coefficient values (Dineshkumar et al., 2019; Torbick et al., 2017). The VH and VV backscatter coefficients have a local minimum value during the transplanting period in all reference paddy rice fields (Figure 2). After transplanting, the VH backscatter coefficients increase as the paddy rice grows and reach a peak at the heading stage (Zhan et al., 2021; Zhang et al., 2018). The VH backscatter coefficients decrease after the rice harvest stage (Phung et al., 2020; Singha et al., 2019; Torbick et al., 2017). Additionally, paddy rice has consistent temporal behavior in the VV/VH ratio and VH. The profiles of the dynamic backscatter coefficients of some land cover types (e.g., water, urban, and forest) are different from those of paddy rice (Figure S3). Therefore, color gradations and the time-series of backscatter coefficients are useful for identifying paddy rice phenology information (Yulianto et al., 2019; Phung et al., 2020; Zhan et al., 2021).

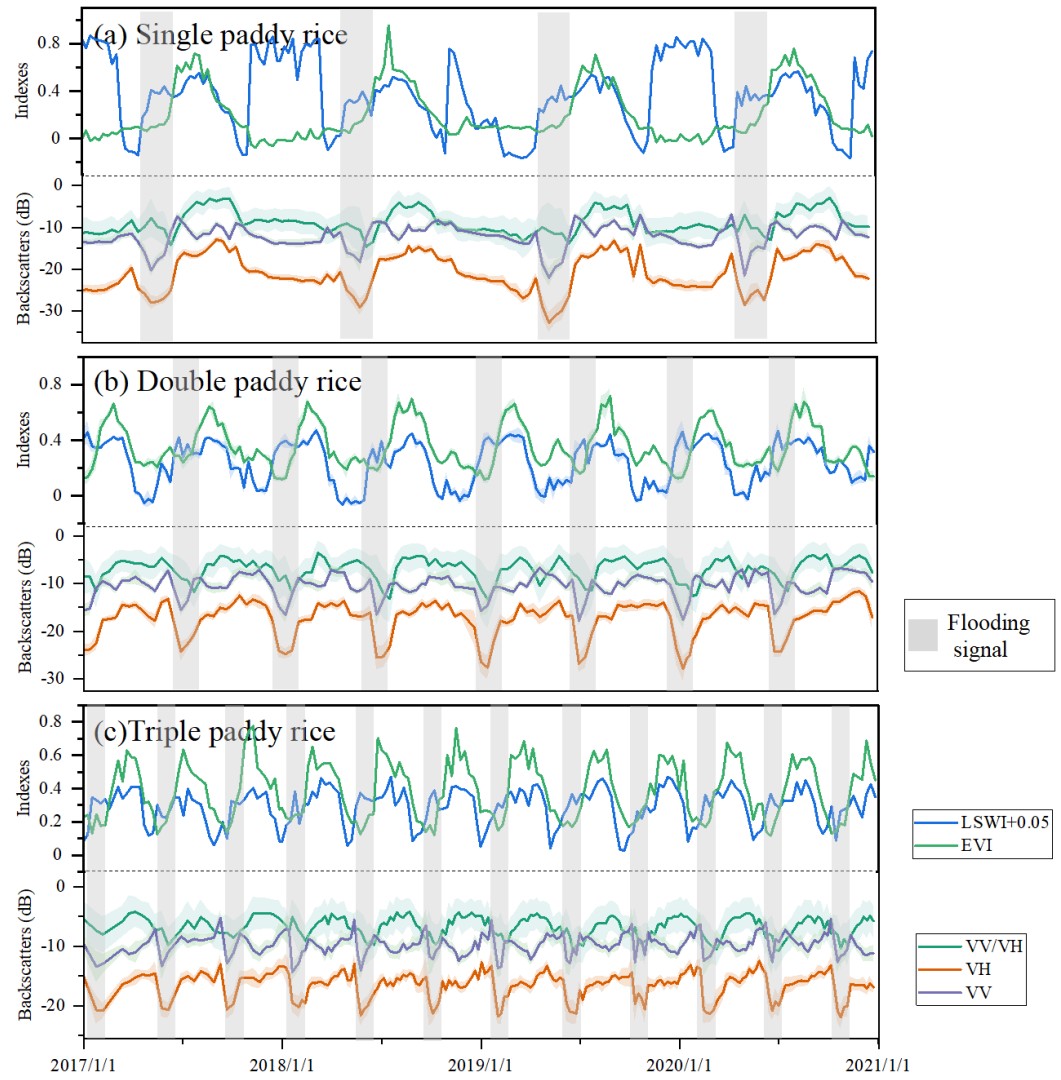

**Figure 2: Temporal profile analysis of EVI, LSWI, VV, VH, and VH/VV from three typical paddy rice sites with different latitudes during 2017–2020: (a) single paddy rice in Northeast China (131.896961°E, 46.804562°N); (b) double paddy rice in the Philippines (120.864242°E, 15.630637°N); and (c) triple paddy rice in Indonesia (116.141157°E, 8.6211997°S). The shaded areas indicate the standard deviation.**

### 2.3.3 Sample blocks collected for extracting phenological parameters

Paddy rice in SE and NE Asia is cultivated using diverse cropping systems because of the climate and other natural conditions (Dong et al., 2016a; Laborte et al., 2017; Nelson et al., 2014; Shew and Ghosh, 2019). With reference to previous studies (Clauss et al., 2016; Gumma et al., 2014; Liu et al., 2020; Phan et al., 2019; Phung et al., 2020), we acquired information about the flooding signal period and length of the growing season for each subzone using sampling-based



information. We selected sample blocks that were distributed over the different rice-growing zones across SE and NE Asia. Each block was a polygon with a radius of 100 m. We collected the sample blocks according to multiple rules (Clauss et al., 180   2016; Dong et al., 2016a; Fikriyah et al., 2019; Singha et al., 2019). First, the time-series of the backscatter coefficients and vegetation index of the mean values from all pixels in each sample block were consistent with the phenological characteristics of paddy rice (Sections 2.3.1 and 2.3.2). Second, the sample blocks were also digitized using Google Earth or Sentinel-1/2 images using visual interpretation referring to previous studies (Dong et al., 2016a; Zhang et al., 2015). Third, we also used existing rice maps and calendar information as complementary information (Laborte et al., 2017; Maclean et al., 185   2013). Note that not all Google Earth or Sentinel-2 images were available throughout SE and NE Asia. We collected a total of 438 sample blocks and 504 sample blocks using the above rules for SE and NE Asia, respectively (Figure 1). These blocks covered most paddy rice fields in the study areas. We generated mean backscatter coefficients and vegetation index time-series profiles for each block. Then, we manually extracted the paddy rice growth and phenological parameters based on the backscatter time-series characteristics. Finally, we obtained the phenological information for each subzone from the 190   sample blocks (Gumma et al., 2014; Liu et al., 2020). Although there may be some limitations in extracting phenological parameters for zones using random samples, it may be one of the most effective approaches currently available (Clauss et al., 2016; Gumma et al., 2014; Han et al., 2021c; Li et al., 2020; Phan et al., 2019; Phung et al., 2020).

### 2.3.4 Algorithm for identifying paddy rice fields

We used a rule-based method to map paddy rice and produce annual paddy rice maps for SE and NE Asia in 2017–2019 at 195   10 m resolution (AsiaRiceMap10m) using the phenological features of paddy rice (Figure 3). The steps for generating the paddy rice maps are as follows:

**Step one.** Detect the flooding area of paddy rice. The key features used to identify paddy rice are the flooding signals in the transplanting phase (Dong and Xiao, 2016). We used LSWI + 0.05 > EVI and LSWI < 0.45 to extract the flooding signals. 200   This rule has been used to successfully map potential paddy rice fields over large areas (Sakamoto et al., 2009; Xiao et al., 2006; Zhou et al., 2016). Because paddy rice flooding signals occur over a short period, non-permanent flooding (e.g., persistent water bodies and fishponds) should be removed (Nelson et al., 2014; Zhang et al., 2015). We removed pixels that had more than 20 composite periods identified as flooding signals during a year.

In addition to the optical MODIS-based LSWI/EVI relationship approach, we also applied the minimum value of VH data in the transplanting stage to identify flooding signals, as suggested in previous studies (Clauss et al., 2018b). VH has a higher sensitivity in paddy rice growth stages than VV polarization (Inoue et al., 2020; Nguyen et al., 2016; Wakabayashi et al., 2019). However, the minimum value of VH in different regions is different because Sentinel-1 data are affected by the incidence angle (ranging from approximately 30° to 45°) (Figure S5) (Phung et al., 2020; Singha et al., 2019; Zhang et al., 210   2018). Currently, it is still challenging to normalize the incidence angle over a large area on the GEE. The VH value of the

water surface changes continuously from -21 dB to -34 dB as the incidence angle changes (Phung et al., 2020). To reduce the incidence angle effect on the Sentinel-1 images, we considered -20 dB as the conservative baseline threshold to achieve the minimum VH value. In previous studies, researchers also proved the effectiveness of this threshold for identifying flooding signals in paddy fields (Clauss et al., 2018b; Nguyen et al., 2015, 2016; Nguyen and Wagner, 2017; Zhang et al., 2018). To
further improve the accuracy of flooding signal extraction, we fine-tuned the baseline threshold of VH for each subzone based on the histogram of sample blocks collected in Section 2.3.3. We considered the pixels that met all the above conditions as flooding signals.

**Step two.** The EVI of paddy rice increased rapidly after the transplanting period because of the increasing numbers of leaves
and biomass (Chen et al., 2012). Therefore, we removed pixels with a maximum EVI value less than 0.4, as suggested in previous studies (Kontgis et al., 2015; Sakamoto et al., 2009).

**Step three.** Moreover, the coefficient of variation (CV) has been proven to be an effective indicator for distinguishing crop types and non-cropland (Huang et al., 2021; Liu et al., 2020; Rose et al., 2021; Whelen and Siqueira, 2018). The VH
backscatter coefficient of crops, particularly of paddy rice, has a larger time-series variation range than non-agricultural land (e.g., urban and water) (Figure S3). Based on this time-series characteristic, paddy rice may be identified from different land surfaces. Therefore, we removed pixels with CV values greater than 0.3 and less than 0.7 calculated using VH during the growth of paddy rice (Rose et al., 2021; Whelen and Siqueira, 2018). We measured the $CV_{VH}$ using the temporal mean (*MEAN*) and standard deviation (*SD*) of the time-series of VH during the paddy rice growth period:

$$CV_{VH} = \frac{SD}{MEAN}. \tag{3}$$

**Step four.** We used the mask with slopes larger than 5° to remove steep terrain; it is unsuitable to plant paddy rice on sloping land (Sun et al., 2009).
**Step five.** We used the PALSAR-based forest map in 2017 as a mask (Wang et al., 2015; Zhang et al., 2017).

**Step six.** Simultaneously, it is challenging to extract paddy rice from wetland because paddy rice and wetland have similar characteristics to flooding signals (Zhang et al., 2015; Zhou et al., 2016). We used the water mask from GlobalLand30 in
2020 referring to the study of Zhang et al (2015) to reduce the misclassification of paddy rice.

**Step seven.** We classified pixels that met all the above rules as paddy rice. Then, we deleted small isolated pixels (connected components less than 12 pixels) to remove the "salt and pepper" effect in the classification




(https://catalog.data.gov/dataset/global-food-security-support-analysis-data-gfsad-cropland-extent-2015-southeast-and-
northe, last access: 2021/06/18).

The single cropping system for paddy rice identification is not ideal because of the difference in paddy rice cultivation time
in some regions of SE Asia (Fikriyah et al., 2019; Shew and Ghosh, 2019). Therefore, we combined all paddy rice fields
identified at different times into the annual map. We applied the improved method to generate the annual paddy rice maps
for SE and NE Asia in 2017–2019. Please note that the method we improved may not extract rice fields (e.g., rain-fed paddy
rice and upland rice) if flooding signals are not available (Xiao et al., 2006; Zhang et al., 2017).

**2.4 Accuracy assessment**

It is challenging to evaluate the accuracy of the classification at continental scales (Xiao et al., 2006; Zhang et al., 2020). We
used two strategies to evaluate the paddy rice maps as accurately as possible. First, we compared the available agricultural
statistics on a subnational level in some countries (Table 1). Referring to the study of Xiao et al. (2006), we calculated the
annual area of paddy rice based on paddy intensity. The paddy intensities of countries in NE Asia, Myanmar, Vietnam, and
the Philippines were 1, 1.4, 2.2, and 2 respectively. Second, we compared the spatial consistency between our classification
results and existing rice maps (Table 1). We used the coefficient of determination ($R^2$) to measure the consistency between
our paddy rice maps, agricultural statistics, and existing products.

$$R^2 = \frac{\left(\sum_{i=1}^{n}(x_i - \bar{x}_i) \times (k_i - \bar{k}_i)\right)^2}{\sum_{i=1}^{n}(x_i - \bar{x}_i)^2 \times \sum_{i=1}^{n}(k_i - \bar{k}_i)^2}, \tag{4}$$


where $n$ is the total number of administrative units, $x_i$ represents the mapped paddy rice areas, $\bar{x}_i$ is the corresponding mean
value, $k_i$ represents the agricultural statistics or areas from existing rice maps, and $\bar{k}_i$ is the corresponding mean value.



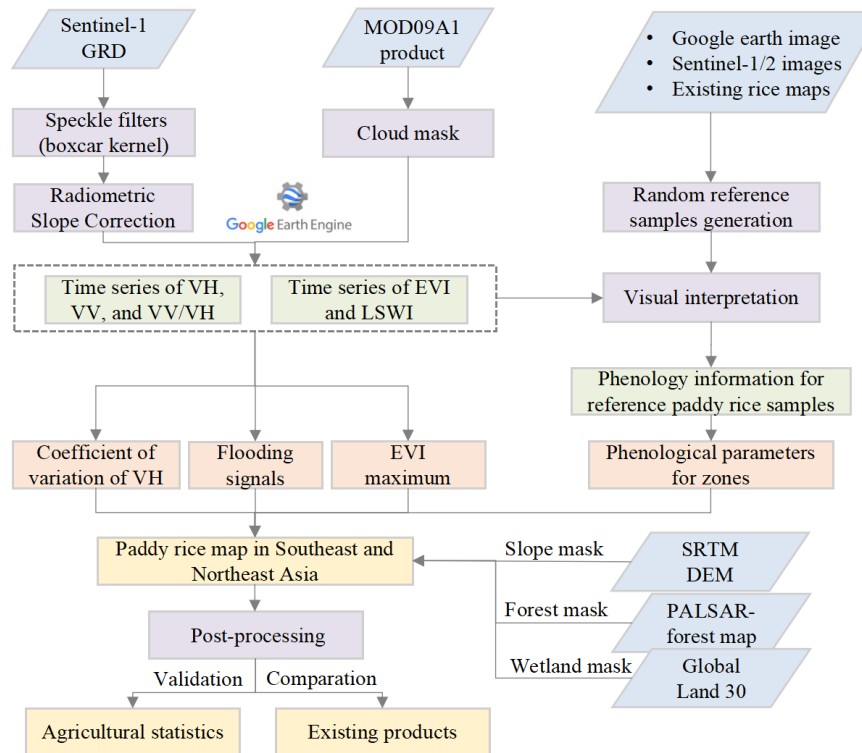

**Figure 3: Flow chart for mapping paddy rice in SE and NE Asia using multiple data.**

## 3 Results

### 3.1 Comparison of the classification with agricultural statistics

The paddy rice maps in SE and NE Asia in 2017–2019 are presented in Figures S6 and S7, respectively. We calculated the annual paddy rice area using the pixel number approach for each administrative unit. The estimated annual rice paddy areas were significantly correlated with the agricultural statistics at subnational levels. The resultant paddy rice maps and the agricultural statistics had relatively high correlations in Northeast China ($R^2$ ranged from 0.82 to 0.89, $p < 0.01$) (Figure 4a). The paddy areas in Changchun, Jilin, and Tonghua city were underestimated. This is mainly because of the lack of available satellite data. When we excluded the three cities, $R^2$ ranged from 0.85 to 0.97, with significant correlations. Additionally, there were significant correlations between the paddy rice maps and agricultural statistics in the Republic of Korea (Figure 4b), but the results underestimated the paddy rice area ($R^2$ ranged from 0.80 to 0.82, $p < 0.01$). The main reason may be that many small rice fields were situated in narrow valleys in the mountains (Dong et al., 2016b; Peng et al., 2011). The correlations were high in most counties in Japan ($R^2$ ranged from 0.89 to 0.93, $p < 0.01$). In Hokkaido, the paddy rice areas were overestimated (Figure 4c), as they were by Zhang et al. (2018). The resultant paddy rice areas were consistent with the agricultural statistics in Myanmar, with $R^2$ ranging from 0.91 to 0.94 (Figure 4d), and in Vietnam, with $R^2$ equal to 0.97 in





the three years (Figure 4e). R$^2$ between the paddy rice maps and the statistical data ranged from 0.81 to 0.87 in the

Philippines, but rice areas in some provinces were underestimated (Figure 4f). Cloud contamination may be a major reason

for the underestimation (Peng et al., 2011; Xiao et al., 2006). The spatial distribution of paddy rice was visually consistent

with that of the higher spatial resolution images in some typical testing regions (Figure S8).

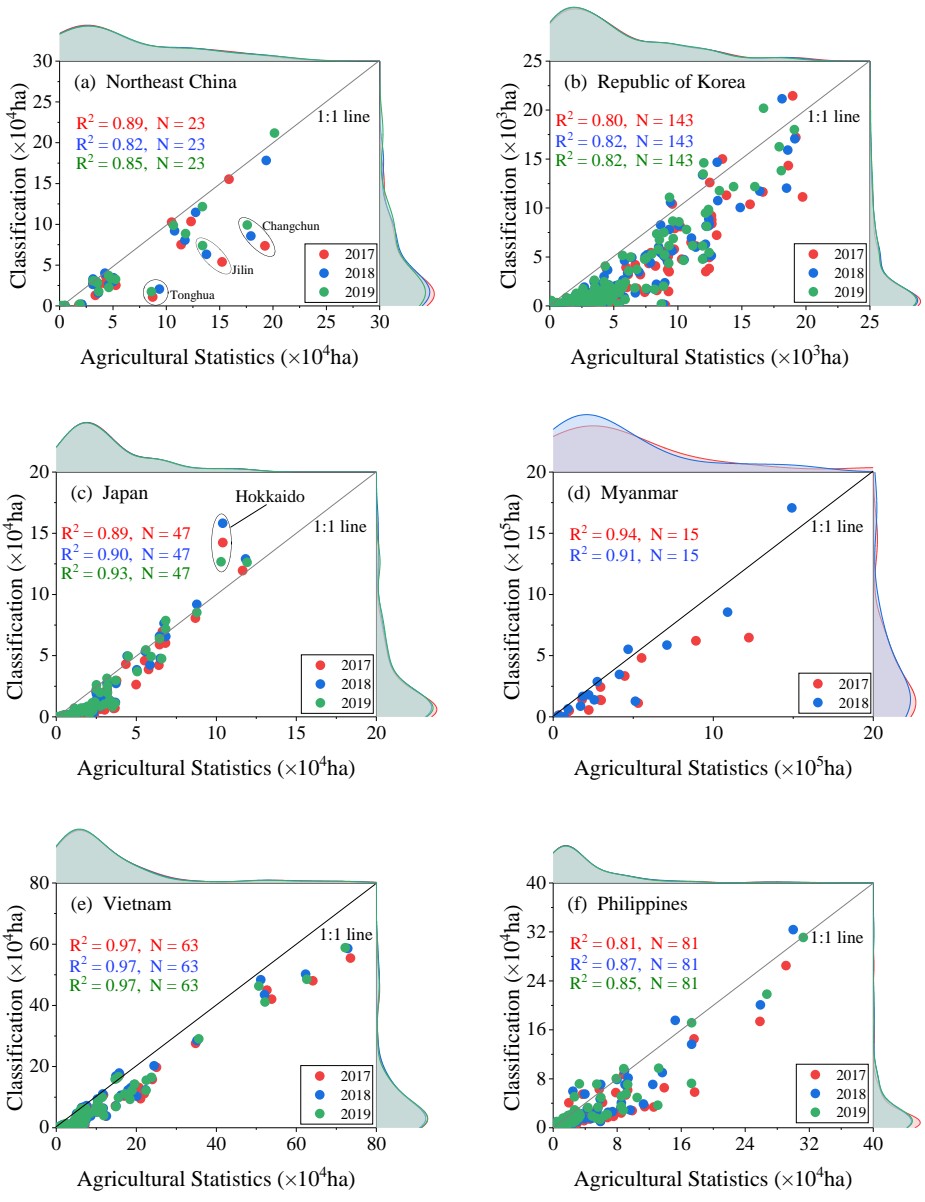


**Figure 4: Comparison of the resultant annual paddy rice areas and the agricultural statistics at the subnational level in different countries from 2017 to 2019.**

Earth System
Science
Data

## 3.2 Comparison of the classification with other annually available rice maps

We further compared the resultant rice maps with existing rice maps at the subnational level. The annually available datasets

included the MODIS-based rice paddy map with 500 m resolution for Northeast China in 2017 and the JAXA-derived rice map with a 10m resolution for Vietnam in 2017 (Section 2.2.6). The paddy rice area statistics from our maps and existing products significantly correlated with $R^2 = 0.99$ (P < 0.01) for Northeast China and $R^2 = 0.93$ (P < 0.01) for Vietnam (Figure 5). We note that the paddy rice area in our maps was smaller than that in the MODIS-based product for Northeast China. The main reason may be that the 500 m resolution MODIS-based paddy rice map had a large number of mixed pixels. Although

the spatial patterns of our maps were consistent with the MODIS-based products (Figures S9 and S10), our maps contained more details with fewer mixed pixels (Figures 9a–f). Additionally, the detailed information at the field scale was consistent for both our maps and the JAXA-derived paddy rice map (Figures 9g–r). Overall, the comparison of the classification with existing products confirmed the reliability of the paddy rice maps that we generated.

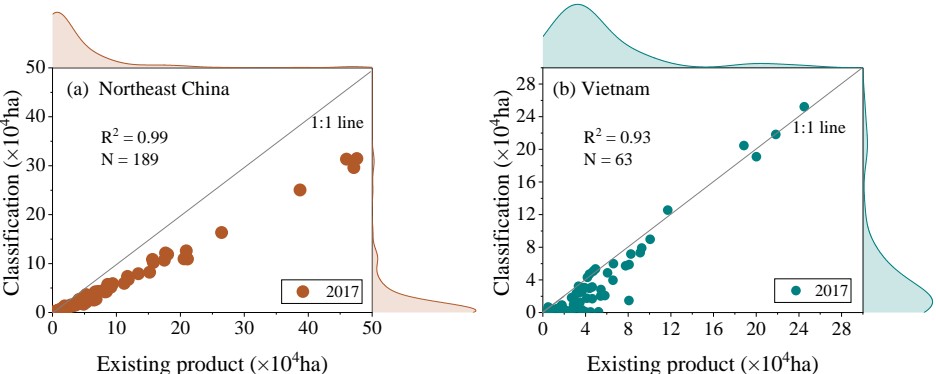

**Figure 5: Comparison of the annual paddy rice area between our classification and existing datasets at the subnational level in Northeast China (a) and Vietnam (b).**

## 3.3 Comparison of the composite paddy rice map and IRRI dataset

The composite paddy rice map is a mosaic of rice planting areas in three years (2017–2019) where rice has been detected in

one or more years. We compared the composite paddy rice areas with IRRI products at the national and subnational levels in SE and NE Asia. The results demonstrated that the correlations between them were significant at both levels ($R^2$ ranged from 0.73 to 0.80) (Figure 6). The paddy rice area based on the IRRI product was higher than our results. The main reason may be that our method reduced the mixed pixels in the paddy rice map and the IRRI product from MODIS overestimated the area, as in previous studies (Figure S11) (Chen et al., 2012; Li et al., 2020; Nelson and Gumma, 2015). The difference between

the two datasets may also be partly caused by the inconsistent epoch composite years in the two datasets. Moreover, although the distribution of paddy rice was consistent between our results and the IRRI product, there were regional differences. Our paddy rice map underestimated the rice area in Thailand (Figure 6a), which may be because the rice planted





in eastern Thailand has no obvious strong flooding signals (Zhang et al., 2020). Despite the various spatial resolutions and different years in the rice paddy data, the inter-comparison verified the accuracy of our dataset.

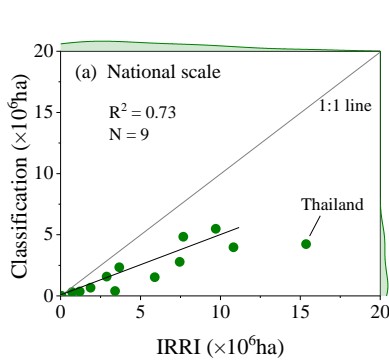
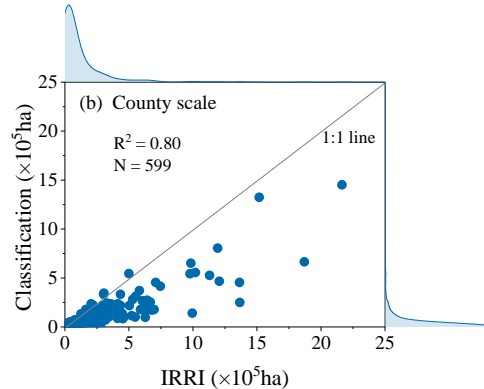


**Figure 6: Comparisons between the paddy rice area in our study and IRRI dataset in SE Asia at the (a) national and (b) county levels.**

## 3.4 Spatial patterns of paddy rice areas

In NE Asia, paddy rice fields are primarily cultivated in the longitude range from 123°E to 134°E and latitude range from 45°N to 48°N (Fig. 7). In Northeast China, paddy rice is mainly cultivated in Heilongjiang (Sanjiang Plain) and Liaoning province. Paddy rice in the Democratic People's Republic of Korea and the Republic of Korea is distributed in the western coastal plains. Some paddy rice fields are located in narrow valleys. Rice fields in Japan are mainly on the coastal alluvial plains.

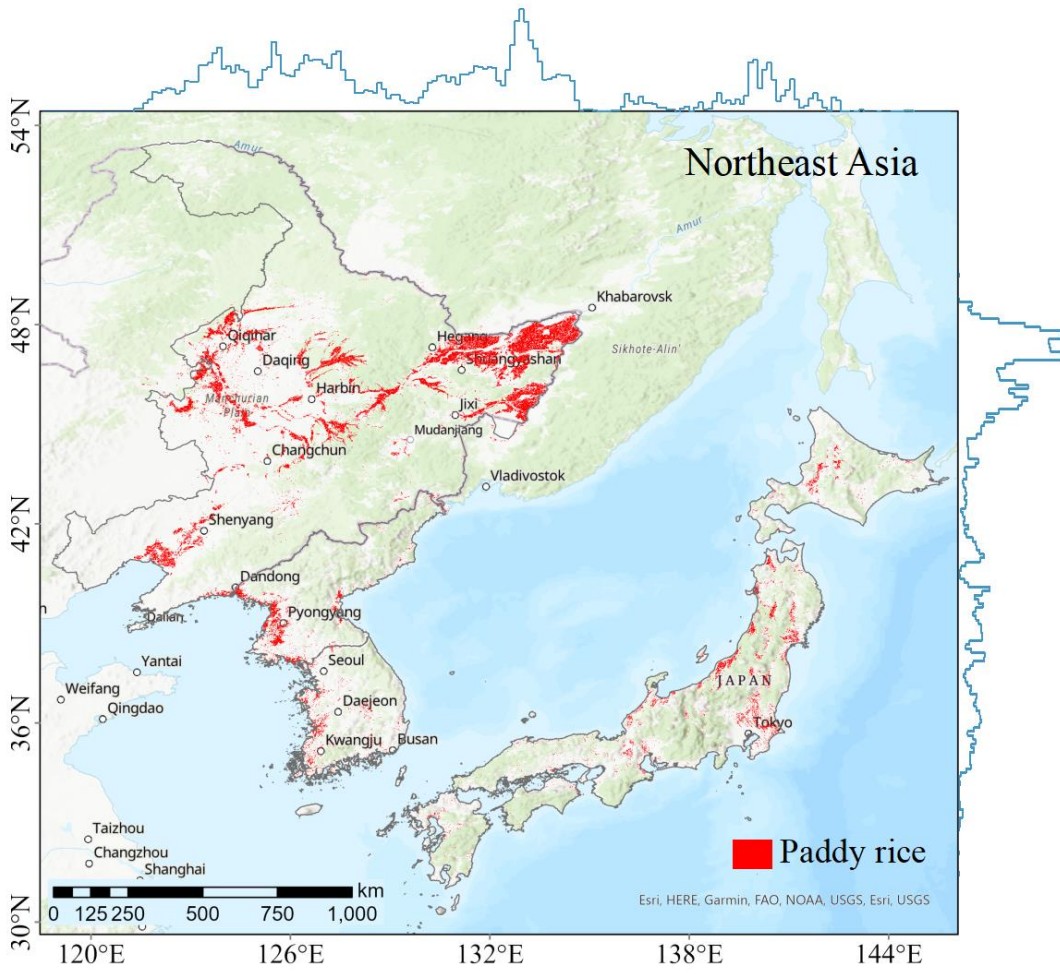

**Figure 7: Spatial distribution of classified composite paddy rice with a 10 m spatial resolution in NE Asia during 2017–2019. The curves represent the distribution of the number of paddy rice pixels' relative change rates along the longitude and latitude gradients.**

Paddy rice is generally planted in the plains and deltas of rivers in SE Asia in the longitude range from 94°E to 106°E and latitude range from 10°N to 21°N (Figure 8). For example, the Mekong Basin and Hong River Delta are typical main rice-growing areas in Vietnam. Paddy rice in the Philippines is mostly grown in the northern plains, and scattered in the Southern Philippines. Most of the rice cultivation in Malaysia is in the northwest corner of the peninsula section.

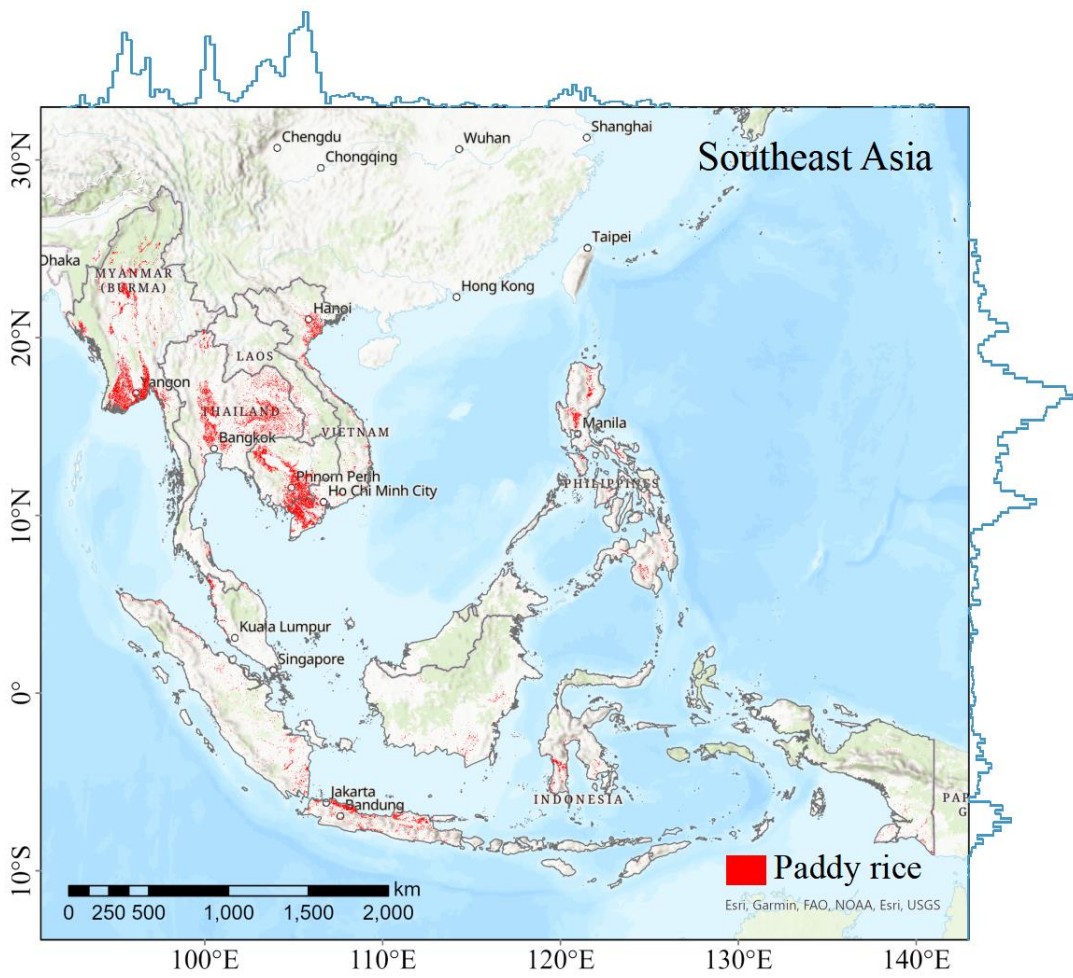

**Figure 8: Spatial distribution of classified composite paddy rice with 10 m spatial resolution in SE Asia during 2017–2019. The curves represent the distribution of the number of paddy rice pixels' relative change rates along the longitude and latitude gradients.**

## 4 Discussion

MODIS data were useful for mapping paddy rice at continental scales using combined EVI and LSWI analysis. Most paddy
rice fields were fragmented in Asia (Li et al., 2020; Lowder et al., 2016). Therefore, it is difficult to solve the intra-class
temporal variability of paddy rice pixels caused by the coarse resolution of 500 m (Dong et al., 2016b; Xiao et al., 2006).
Mixed pixels may cause an overestimation of the rice cultivation areas (Nelson and Gumma, 2015). We improved the
MODIS-based approach by incorporating Sentinel-1 data, and used the approach to identify paddy rice fields in SE and NE
Asia for 2017–2019. Reducing the mixed pixel problem is the key point of the improved paddy rice mapping method.
Compared with the paddy rice maps acquired from existing MODIS-based products, our classification provides more

information about field details with a higher spatial resolution (10 m) (Figure 9). Therefore, the integration of MODIS and Sentinel-1 data makes it possible to improve the accuracy of mapping paddy rice at continental scales.

**Figure 9: Visual comparison of our paddy rice maps and existing products in typical regions in 2017: (a–c, g–i, m–o) classification using our method; find the example data for (a), (g) and (m) here (https://doi.org/10.17632/cnc3tkbwcm.1, example01-03); (d–f) MODIS-based paddy rice fields; and (j–l, p–r) JAXA-derived rice fields.**



Although our paddy rice maps are consistent with existing products, some uncertainty sources still affect the mapping results. First, identifying small paddy rice fields in hilly regions is challenging for MODIS data, which will lead to an underestimation of the area of paddy rice fields (Dong and Xiao, 2016; Zhang et al., 2015). For example, the rice planting

area is smaller than the agricultural statistics in the mountainous provinces of the Republic of Korea (Figure 10). Second, the classification method relies on rice paddies containing irrigation water during transplanting stages. Therefore, rain-fed paddy rice and upland rice may not be detected because of the unavailability of flooding signals (Zhang et al., 2017). The main reason for the underestimation of the rice area in eastern Thailand may be that the flooding signal of rice was not detected, which has also been mentioned in previous studies (Bridhikitti and Overcamp, 2012; Guo et al., 2019; Zhang et al., 2020).

Third, although MODIS data with a high temporal resolution was used in our method, the accuracy of rice maps is still affected by cloud contamination (Dong and Xiao, 2016). Fourth, missing observations in Sentinel-1 data would lead to noteworthy omission errors. Fifth, both the thresholds of different indicators and phenological information extracted by sample blocks may affect the accuracy of the classification (Dirgahayu and Parsa, 2019; Jeong et al., 2012; Li et al., 2020; Yeom et al., 2021). Sixth, other land cover products used in this study may also affect the accuracy of the classification.

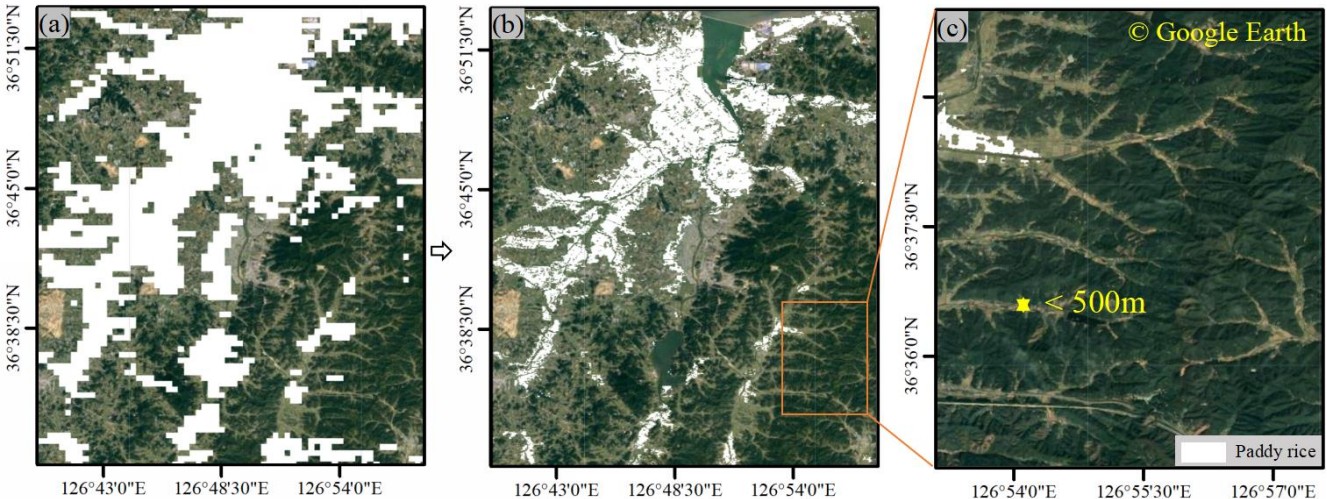


**Figure 10: Estimated distribution of paddy rice in 2017 in mountainous regions in South Korea: (a) flooding signal based on MODIS indices; (b) paddy rice map generated by our method; find the example data for (b) here (https://doi.org/10.17632/cnc3tkbwcm.1, example04); and (c) mountainous landscapes acquired from © Google Earth.**




## 5 Data availability

The datasets of the paddy rice maps for SE and NE Asia from 2017 to 2019 are available on Mendeley Data. Find small
example here data (https://doi.org/10.17632/cnc3tkbwcm.1) (Han et al., 2021b), or download the full product here
(https://doi.org/10.17632/j34b3jsvr9.1) (Han et al., 2021a). The spatial reference system of the datasets is EPSG: 4326. We
encourage users to validate this dataset independently. Please note that some small islands are not classified, and thus data
for these areas are not available.

## 6 Conclusions

We constructed a paddy rice map database for SE and NE Asia for three years (2017–2019) at a 10 m spatial resolution
(AsiaRiceMap10m) by integrating MODIS and Sentinel-1 data. The paddy rice planting areas in our database were
significantly correlated with those from the official statistics. The distribution of paddy rice in the maps was consistent with
existing data products. Additionally, our method reduced the effects of mixed pixels and provided more detailed spatial
information than MODIS-based paddy rice maps. We demonstrated that multi-sensor data integration has the advantages of
improving the spatial resolution of rice maps and reducing mixed pixels. To summarize, we provided more accurate paddy
rice maps at continental scales using the improved method for paddy rice mapping.

## Author contributions

Conceptualization, ZZ and JH. Data curation, JH, YL, and JC. Formal analysis, JH and ZZ. Methodology, JH and YL.
Validation, JH, JC and LZ. Visualization, JH. Writing – original draft preparation, JH, ZZ, YL, and JC. Writing – review &
editing, JH, ZZ, YL, JC, LZ, FC, HZ, and JZ.

## Declaration of competing interest
The authors declare no conflict of interest.


## Acknowledgments

This work was supported by the National Natural Science Foundation of China (project no. 42061144003). We thank the
editors and anonymous reviewers for their valuable comments. We thank Maxine Garcia, PhD, from Liwen Bianji (Edanz)
(www.liwenbianji.cn/, last access: 2021/08/23) for editing the English text of a draft of this manuscript.






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
