# Peer review of "NESEA-Rice10: high-resolution annual paddy rice maps for Northeast and Southeast Asia from 2017 to 2019"

_Earth System Science Data, 2021_

## Author Response (AR1)

Dear Editor and Reviewer # 1:

Thanks for your careful reviewing our manuscript. Those comments are all valuable and helpful for revising and improving our researches. We have studied comments carefully and have made a correction which we hope meets with approval. The point-to-point responses to your comments are listed below in **blue.**

**Point:** I suggest revising the name of the data set in the title since the dataset proposed in this work covers Southeast Asia and Northeast Asia. You may consider covering more countries or regions of Asia in the future.

**Response:** We have revised the name of the dataset to NESEA-Rice10. Also, we have revised the title in the revised manuscript and the revised supplementary material (title: NESEA-Rice10: high-resolution annual paddy rice maps for Northeast and Southeast Asia from 2017 to 2019).

**Point:** I noticed that the author showed the time series of spectral indices and backscattering coefficients of typical rice planting areas in the manuscript. I suggest that the authors add some figures of the spectral index time series for other land types in the supplement.

**Response:** We have added spectral index time series for other land types in the supplementary material. Also, we have added the description of the

time series for other land types. For more details, please see Figure S3 in the revised supplementary material.

**Point:** Please explain the meaning of the curve distribution around the Figures 4-6 in their captions.

**Response:** We have explained the meaning of the distribution of the curves around it in the caption of Figure 4-6. Also, The marginal kernel density plot above or to the right of each scatter plot shows the distribution of the data in one dimension. Please see lines 294-295, 309-310, 326-327 for more details in the revised manuscript.

**Point:** I suggest that the authors should add some descriptions about the future development of the rice map data set in the discussion section.

**Response:** Many thanks for your constructive comment. We have added a description of potential applications and future developments of the paddy rice dataset in the Discussion section. Please see lines 386-405 for more details in the revised manuscript.

"Under the combined effects of climate change and human activities, such as frequent extreme disasters, population growth, and urban expansion, paddy rice, knowing the spatial distribution of paddy rice is important for food security. The potential applications of the dataset include: (1) improving paddy rice yield prediction accuracy. Crop masks are the basis

for paddy rice yield prediction. Previous studies have demonstrated that the accuracy of crop masks affects the accuracy of yield prediction (Liu et al., 2019a; Zhang et al., 2019); (2) assessing damage to agriculture from extreme hazards. Floods are one of the major natural disasters in Southeast Asia. High-resolution paddy rice maps will improve the accuracy of the area and yield loss estimates for flooded farmland (Phan et al., 2019); (3) estimating green-house relevant methane emissions. Paddy rice is an important source of methane in the atmosphere (Redeker et al., 2000). Accurate paddy rice maps and crop intensity maps facilitate the estimation of methane emissions (Zhang et al., 2020); In addition, paddy maps are helpful in making land-use decisions for the government, etc.

Recently, as more Sentinel-2 images with higher resolutions are available, combining Sentinel-2 and other satellite images will improve the temporal resolution of the data. For example, the Harmonized Landsat and Sentinel-2 (HLS) project provide images with 2–3 days at 30m spatial resolution by combining Landsat 8 satellite and Sentinel-2 satellite (https://hls.gsfc.nasa.gov/, last access: 2021/11/5). Zhang et al. (2021) mapped the global cropping intensity with a high spatial resolution by integrating Sentinel-2, Landsat 8, and MODIS satellites (Zhang et al., 2021). Therefore, combining multi-source remote sensing data provide opportunities for global rice mapping in the future. Also, the increasing number of Planet satellite images at higher resolutions (3 - 5 meter) could

further improve the accuracy of the paddy rice map in the future (https://www.planet.com/, last access: 2021/11/5). To further improve the accuracy of paddy rice map products, more accurate information on cropland and forest masks, and crop calendars will need to be developed in the future."

**Point:** The map boundaries of Figure 9m and Figure 9p are not the same, please revise Figure 9p.

**Response:** Thank you very much for your careful review. We have modified the map boundaries of Figure 9m and Figure 9p. Please see Figure 9 in the revised manuscript for more details.

Dear Editor and Reviewer # 2:

We appreciate your insightful comments on our paper. The comments offered have been immensely helpful. We have responded to every question, indicating exactly how we addressed each concern or problem and describing the changes we have made. The revisions have been approved by all authors. The point-to-point responses to your comments are listed below in **blue.**

**Point:** The title should be changed since the dataset only covers Southeast Asia and Northeast Asia, rather the whold region of Asia.

**Response:** We have revised the title in the revised manuscript and the revised supplementary material (title: NESEA-Rice10: high-resolution annual paddy rice maps for Northeast and Southeast Asia from 2017 to 2019).

**Point:** The description of the data preprocessing process of the spectral indices (EVI and LSWI) in chapter 2.3.1 needs to be more clear. I want to know if the authors used some filtering method to smooth the time series.

**Response:** Sorry for the unclear expression. In this study, we did not smooth the EVI and LSWI time series. LSWI changes under different dry and wet conditions and smoothing the EVI and LWSI time series may

eliminate the true paddy rice flood signal. Therefore, we did not reconstruct the EVI and LSWI datasets. We have added a description in the revised manuscript (lines 159-161).

**Point:** The authors need to clearly explain the reason for the coefficient of variation (CV) in chapter 2.3.4.

**Response:** Thank you very much for your careful review. The threshold of the coefficient of variation (CV) was determined from the histogram of the paddy rice sample blocks in the study area. We added the histogram for determining thresholds of CV in the revised manuscript (Figure S7) and added descriptions in the revised manuscript (lines 234-235).

**Point:** I noticed that the authors have explained that the availability of satellite imagery will affect the accuracy of the data set in the discussion section, which is good. If the authors add the availability map of Sentinel-1 images in the study area (Southeast Asia and Northeast Asia) in the manuscript, it will help users better understand their data set.

**Response:** We greatly appreciate your valuable comment. We have added the spatial distribution map of good-quality observation numbers from 2017 to 2019 for MOD09A1 and Sentinel-1 images in Northeast Asia and Northeast Asia. For more details, please see Figure 11 in the revised manuscript.

**Point:** The dataset is organized as an archive of GeoTIFF files in the Mendeley Data. I suggest that the authors reorganize the structure of the dataset in a geographic grid (e.g., 5°×5°). Also, rename each file among the data set. e.g., filename 105E20N2019.tif.

**Response:** Thanks for your valuable comment. We have reorganized our data storage structure in a geographic grid (5°×5°) to make it easier for users to use. Also, we have renamed the data for a better user understanding of the product. Please see the Data availability section in the revised manuscript for more details.

**Point:** The legend of the paddy rice in Figure 10 is white, which is the same as the background of the paper. Please change the legend of the paddy rice, such as red.

**Response:** We have modified the legend in Figure 10.